# Surface Modification of Zinc Ferrite with Titanium to Be a Photo-Active Catalyst in Commercial LED Light

**Doaa F. Baamer** [1] **and Islam Hamdy Abd El Maksod** [2,*]

[1] Chemistry Department, Faculty of Science, King Abdulaziz University, Jeddah P.O. Box 42805, Saudi Arabia; doody.84@hotmail.com
[2] Physical Chemistry Department, National Research Centre (NRC), Cairo 12622, Egypt
* Correspondence: islam_9000@yahoo.com

**Abstract:** Titanium-doped zinc ferrite was used as a photo catalyst for breaking down C-C and C-H bonds of methylene blue dye as a model for the decomposition of organic pollutants. Different concentrations of Ti were used to impede into the spinel structure of zinc ferrite by in situ addition during the preparation. Different characterization techniques were used to characterize the prepared materials including the deep analysis of the electronic spectra, which proved the surface modification of ferrite due to the Ti doping. In addition, we make a comparison study of photo degradation using ordinary UV irradiation and commercial LED light irradiation, which gives very promising results. A correlation between the structure and the photo catalytic behavior of the materials is assigned.

**Keywords:** zinc ferrite; spinel structure; Ti doping; photo-active centers; LED light; photodegrading

## 1. Introduction

The ferrites materials are those compounds composed of the spinel structure of $MFe_2O_4$, where M is the divalent cation [1,2]. The ferrite materials have a wide application as a ceramic material due to their high magnetic properties. They were used a long time ago in many electronic devices and also as catalysts and support for many catalysts to acquire the easily magnetic separation behavior [3–9]. Although the ferrite materials were used as catalyst for many reactions, there is very limited application for their use as photo catalysts. This may be due to the bandgap values of the spinel structure, which do not accommodate the requirements of a photo catalyst. However, the use of a supported photo catalyst on ferrite may be considered as a good solution or include some divalent cations in the spinel structure to accommodate the bandgap values [10–13]. Another solution may be given by the use of doping the spinel structure with some foreign cations, such as Al, which makes a tremendous change in ferrite behavior [14]. Many papers used ferrite as a support for the $TiO_2$ photo catalyst [15–20]. In these papers, the adding of the Ti precursors on already prepared ferrite materials is performed. The resultant composite material consists of two separate phases of $TiO_2$ and ferrites. Although the good results could be obtained from this style of preparation, it takes a long time for photo degradation due to the limitation arises of the amount of loaded $TiO_2$ without losing the magnetic behavior of ferrites. Instead, in this paper we will try to modify the internal structure and the electronic behavior of the ferrite by incorporation of Ti into its spinel structure by in situ doping of Ti into the synthesizing mixture of ferrites, which is expected lead to a high performance of ferrites as a photo catalyst without any losing of magnetic behavior. In addition, we tried to use commercial LED light to test the prepared catalyst. The prepared photo catalyst is expected to enhance the degradation process of C-C and C-H bonds in the dye Skelton backbone.

The dye is considered to be from the most pollution sources in industry. Its removal is the main task for controlling its pollution. In addition, health impacts may cause very

harmful effects on the human and ecological system. The removal process includes adsorption and degradation. The adsorption method, however, has many drawbacks regarding the environmental issues that arise from how to rid the adsorbed dye. Hence, a degradation process may be the best choice. The degradation will include chemical degradation and photo degradation. The chemical degradation, however, may complicate the environmental problems due to the high oxidizing agent used, which affects all ecological and aqua life systems. The photo degradation will be the best choice for which to decompose the dye without any additional chemical wastes.

## 2. Results and Discussion

### 2.1. X-ray Diffraction (XRD)

Figure 1 showed XRD diffractograms of different samples prepared. It could be observed easily that the undoped zinc ferrite sample (0) showed a pure phase of only zinc ferrite as compared to PDF cards. The analysis of crystalline lattice parameters (Table 1) showed also that this phase is very similar to the reference PDF card showing the cubic crystal system. Moreover, the addition of titanium to the doped titanium samples showed a remarkable change in lattice parameters transferring it into a tetragonal system (Table 1), indicating that, doubtlessly, titanium enters the lattice of zinc ferrite. In other words, the change in the lattice parameters means that Ti substitutes metal in the spinel structure of ferrite, and due to difference in atomic radii, the difference in the lattice parameters is observed. In addition, a very small amount of ZnO was observed to be formed by the entrance of titanium. Hence, the Zn ions occupy the tetrahedral sites of the spinel structure of ferrite and the incorporation of Ti is accompanied by the appearance of a separate ZnO phase; it is not difficult to conclude that the titanium substituted Zn in the tetrahedral position, resulting in the rippling of Zn ions out of the spinel lattice, which clearly changes the lattice parameters (Scheme 1).

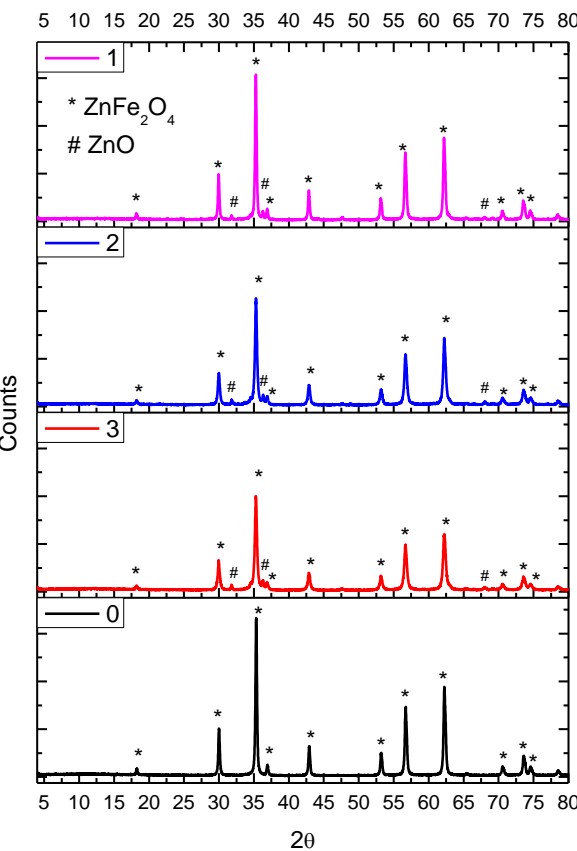

**Figure 1.** XRD diffraction pattern of different prepared samples.

**Table 1.** XRD lattice analysis of different samples.

| Crystal Lattice Parameters | | | | | |
|---|---|---|---|---|---|
| Sample | a | b | c | Crystal System | Crytallite Size (nm) |
| Refrence (ZnFe$_2$O$_4$) PDf no p221012 | 8.441 | 8.441 | 8.441 | Cubic (Fd-3m) | |
| Sample 0 | 8.4357 | 8.4357 | 8.4357 | Cubic (Fd-3m) | 57.33 |
| Sample 3 | 9.739 | 9.739 | 9.377 | Tetragonal (I 41/a c d) | 48.36 |
| Sample 2 | 7.776 | 7.776 | 10.436 | Tetragonal (I 41/a c d) | 53.31 |
| Sample 1 | 8.701 | 8.701 | 7.974 | Tetragonal (I 41/a c d) | 78.7 |

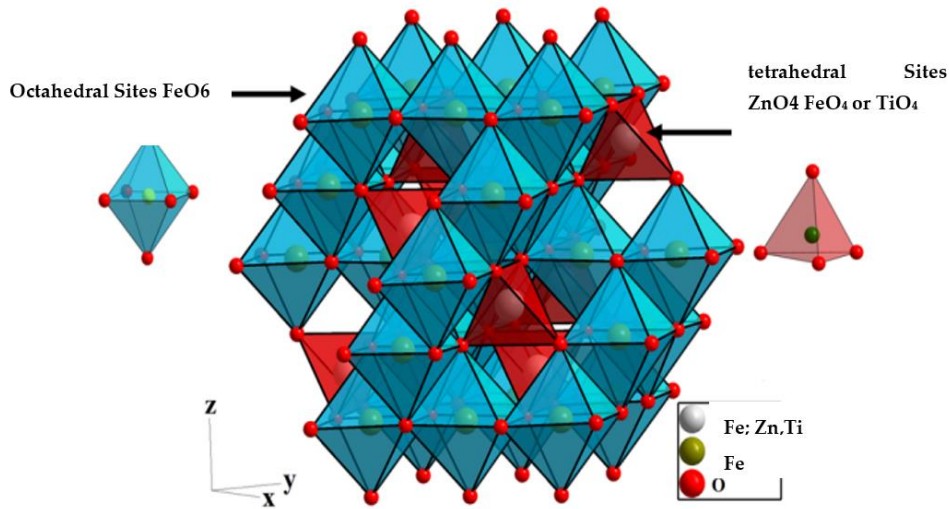

**Scheme 1.** Schematic diagram of the spinel structure of zinc ferrite. Zn and Ti occupy the tetrahedral position.

*2.2. Diffuse Reflectance*

DR UV–vis spectral data were gained using the UV-Vis spectrometer Jasco v770. The detailed experimental procedure can be described in the reference [19]. Band gap energy values of all the samples were calculated using the Kubelka–Munk method (details of this method are found in our previous work [21]). The Kubelka–Munk factor (K) was determined by the following equation:

$$K = (1 - R)^2/2R$$

where R is the % reflectance. The wavelengths (nm) were translated into energies (E) and a plot was drawn between $(K * E)^{0.5}$ and E to obtain a curve. The bandgap energy (eV) was obtained as the intersection point of the two slopes in the curve.

The curves and the values of the band gabs of all samples under investigation are given in Figure 2. From this figure, it can be observed that the band gap values are in the range of the zinc ferrite values (1.85–190) [22]. In order to deep analyze the electronic spectra of the investigated samples, we performed a deconvolution analysis of the peaks (Figure 3 and Table 2).

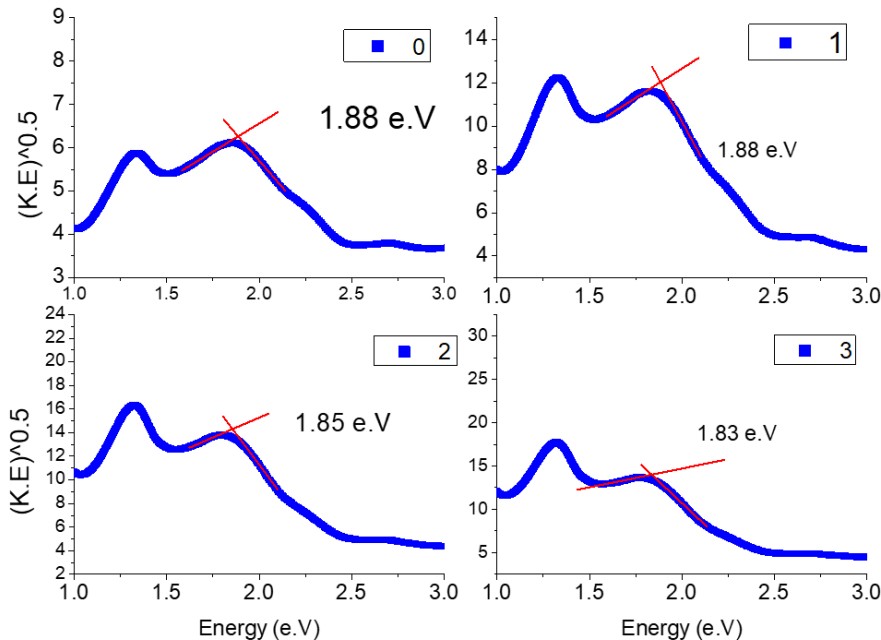

**Figure 2.** Kubelka–Munk figures for calculation the band gab values.

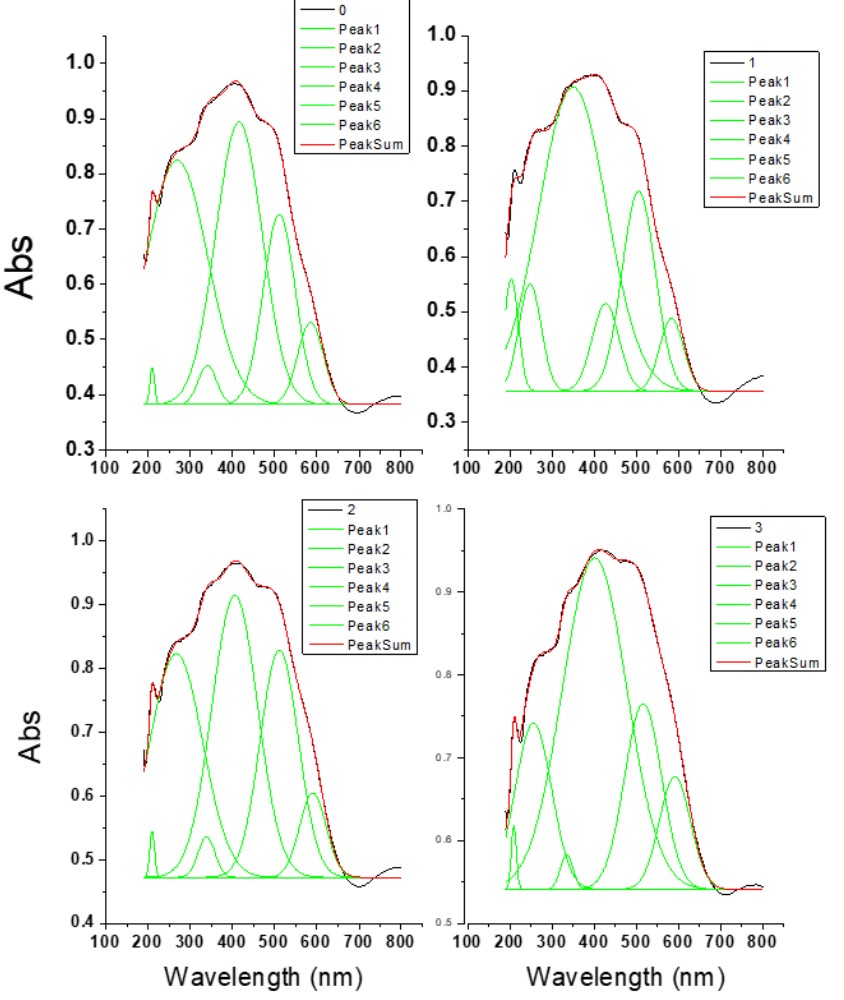

**Figure 3.** Deconvolution analysis of diffuse reflectance spectra of samples under investigation.

**Table 2.** Data for deconvolution analysis of peaks.

| Sample 0 | | | | | Sample 2 | | | | |
|---|---|---|---|---|---|---|---|---|---|
| | **Area** | **Center** | **Width** | **Height** | | **Area** | **Center** | **Width** | **Height** |
| 1.00 | 0.93 | 209.65 | 11.01 | 0.07 | 1.00 | 1.03 | 209.53 | 11.36 | 0.07 |
| 2.00 | 80.33 | 268.71 | 144.74 | 0.44 | 2.00 | 55.17 | 266.76 | 125.36 | 0.35 |
| 3.00 | 3.92 | 340.65 | 44.02 | 0.07 | 3.00 | 3.24 | 338.42 | 40.57 | 0.06 |
| 4.00 | 70.70 | 415.44 | 110.29 | 0.51 | 4.00 | 60.35 | 405.54 | 108.78 | 0.44 |
| 5.00 | 32.92 | 511.08 | 76.25 | 0.34 | 5.00 | 40.57 | 511.27 | 90.64 | 0.36 |
| 6.00 | 11.42 | 585.27 | 61.11 | 0.15 | 6.00 | 10.43 | 590.73 | 62.79 | 0.13 |
| sample 1 | | | | | sample 3 | | | | |
| | **Area** | **Center** | **Width** | **Height** | | **Area** | **Center** | **Width** | **Height** |
| 1.00 | 8.56 | 203.53 | 33.48 | 0.20 | 1.00 | 1.17 | 209.48 | 12.12 | 0.08 |
| 2.00 | 12.98 | 247.85 | 53.20 | 0.19 | 2.00 | 21.75 | 255.89 | 86.22 | 0.20 |
| 3.00 | 110.81 | 349.55 | 160.49 | 0.55 | 3.00 | 1.61 | 334.66 | 30.35 | 0.04 |
| 4.00 | 12.74 | 427.08 | 63.77 | 0.16 | 4.00 | 76.74 | 401.16 | 152.98 | 0.40 |
| 5.00 | 35.78 | 505.44 | 78.76 | 0.36 | 5.00 | 24.82 | 516.24 | 88.30 | 0.22 |
| 6.00 | 9.40 | 583.75 | 56.56 | 0.13 | 6.00 | 12.63 | 591.40 | 73.84 | 0.14 |

This deconvolution analysis could be used successfully to more or less quantify the electronic transition between valence gap and conduction band and give a clear sight on what exactly happen at these transitions.

By deep analysis of the deconvoluted peaks of the electronic spectra of zinc ferrites samples, it could be observed that all spectra are in common in most peaks; however, two distinguished peaks appear as one at 395, which is attributed to Zn-O lattice defects [22], and the other at 345, which is attributed to nano particles of TiO$_2$.

Peaks at ~395 nm just appear in sample 2 and 3. This could be attributed to the intrinsic point defects in the crystal lattice of Zn-O in the spinel structure of ferrite as interstitial zinc ions and zinc vacancy [22] while peaks at ~345 nm are attributed to the presence of TiO$_2$ [23]. These peaks appeared only in sample 1, indicating that Ti in sample 1 is agglomerated on itself, forming TiO$_2$ nano particles entering the lattice of ferrite without significant replacement of zinc ions, as evidenced from crystal lattice analysis (Table 2), which showed the large crystal size of ferrite and large deviation in lattice parameters in sample 1. Moreover, Ti in sample 2 and 3 enters the lattice, changing the defect positions of the zinc ferrites.

Moreover, the presence of different deconvoluted peaks in the DR spectrum indicates that we have more than one transition rather than the apparent band gap transition, which could explain the photo activity of different sites with respect to different irradiations.

### 2.3. SEM Images and EDX Analysis

SEM images of all investigated zinc ferrites samples are given in Figure 4. It could be observed that all samples showed clear crystals of ferrites, which can be analyzed in the image and calculated in Figure 5. These values are very similar to those obtained from Table 2 using the Scherer equation analyzed before from XRD pattern. Except for the zero sample, for which the analysis of the SEM image showed larger values. This could be explained by the fact that the pure zinc ferrite attains more magnetism, which shows much agglomeration in the SEM image that prevents analysis to be accurate; while in other samples, the presence of Ti makes what is called solid dilution decrease more or less magnetism, which makes it is possible to analyze the crystals accurately [24]. In addition,

Table 3 showed the EDX analysis of all samples accompanied with the calculations of surface excess concentration, which could be given in the equation:

Percent surface excess concentration = ((actual concentration of element given by EDX- theoretical calculated amount from synthesizing mixture)/(theoretical calculated amount from synthesizing mixture)) × 100.

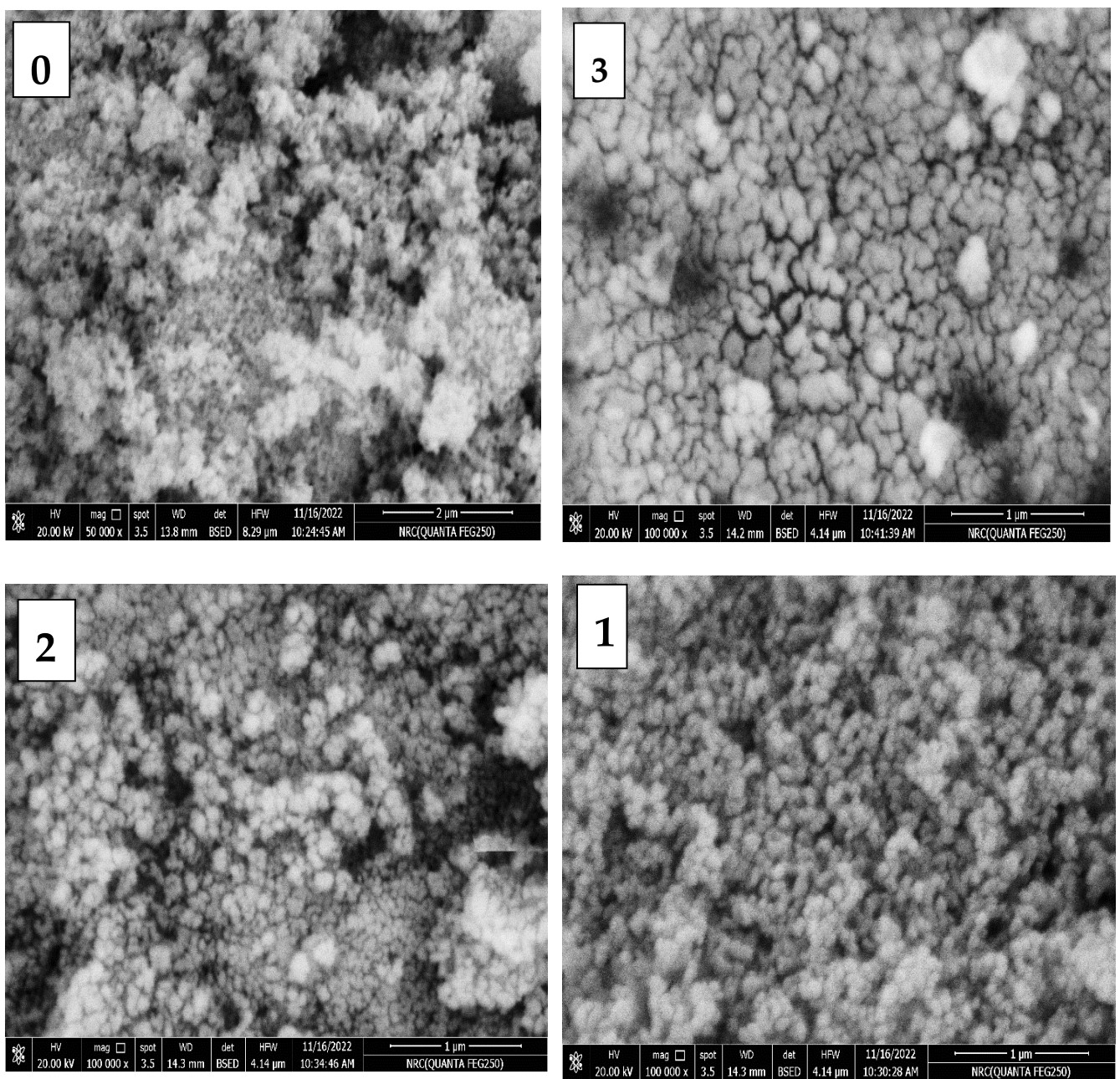

**Figure 4.** SEM of all samples under investigation.

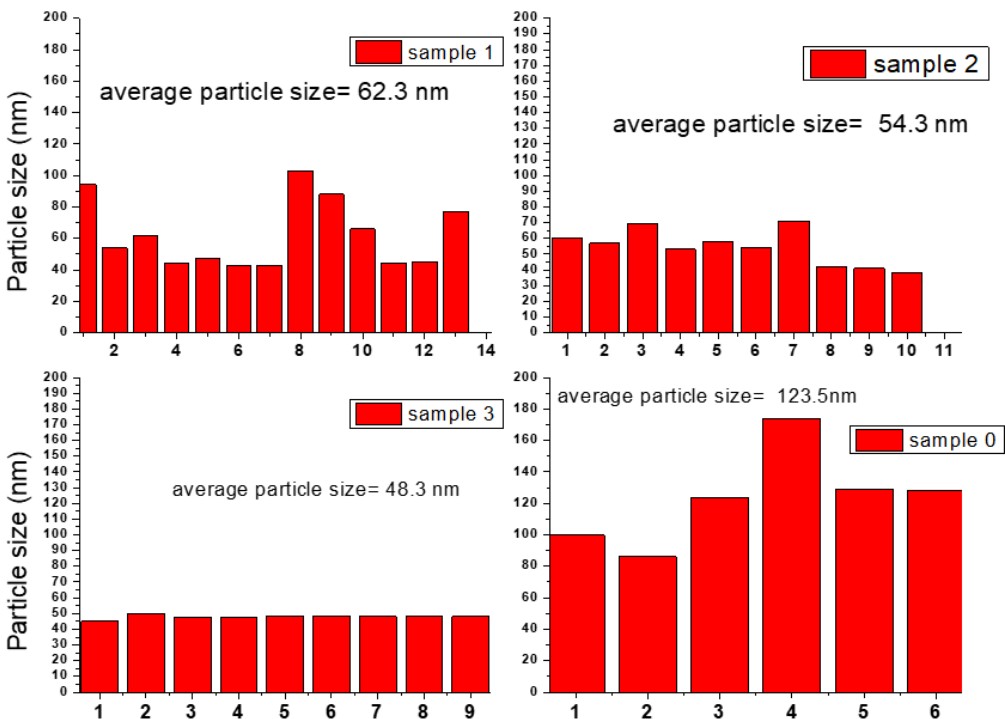

**Figure 5.** Particle size distribution curves of SEM images.

**Table 3.** EDX analysis of investigated samples and surface excess concentrations.

| | Theoretical | | | |
|---|---|---|---|---|
| | Ti | Zn | Fe | O |
| **0** | 0 | 14.28 | 28.5 | 57.1 |
| **1** | 1.759 | 13.53 | 27.06 | 57.6 |
| **2** | 1.11 | 13.8 | 27.6 | 57.4 |
| **3** | 0.699 | 13.98 | 27.97 | 57.3 |
| **Actual** | | | | |
| | Ti | Zn | Fe | O |
| **0** | 0 | 12.63 | 27.52 | 59.85 |
| **1** | 1.9 | 13.53 | 23.97 | 60.59 |
| **2** | 2.09 | 14.69 | 26.06 | 57.15 |
| **3** | 1.1 | 14.42 | 28.37 | 56.1 |
| | Surface excess % | | | |
| | Ti | Zn | Fe | O |
| **0** | 0 | −11.5546 | −3.4386 | 4.816112 |
| **1** | 8.015918 | 0 | −11.4191 | 5.190972 |
| **2** | 88.28829 | 6.449275 | −5.57971 | −0.43554 |
| **3** | 57.36767 | 3.147353 | 1.430104 | −2.09424 |

From this table, it can be observed that the zinc in zero samples showed −ve value in surface excess concentration, indicating that zinc in pure samples tends to be in the bulk rather than on the surface, while it gave +ve values in samples 2 and 3, which contain less amounts of Ti. Sample 1, however, showed zero surface excess. In addition, Ti surface excess showed higher values in samples 2 and 3, while there were only low +ve values in the case of sample 1. The −ve value in surface excess concentration means that the

element exists on the surface in less of an amount than that theoretically assumed of equal distribution between bulk and surface. While +ve values reflect that most of element concentration is concentrated on the surface rather than bulk.

From these data, it could be concluded that Ti doping modified the surface zinc in ferrite samples and accumulated on the surface in low doped amounts (samples 2, 3), while for sample 1, with a larger titanium amount, the Ti tends to enter the crystal of zinc ferrite as an inclusion in the form of $TiO_2$ nano particles, which was previously evidenced from XRD analysis.

### 2.4. Textural Properties of Different Investigated Samples

Figure 6 shows the adsorption isotherm of different samples. From this figure, it can be observed that all samples followed the type II isotherm. Moreover, a hysteresis loop was observed only in sample 3, with a minimum amount of Ti doping. The surface area and pore size were given in Table 4. From this table, the surface area measurements are found to be in the range of 15–30 $m^2/g$, which is classified as very low surface area. In addition, a remarkable decrease in pore size in sample 3 (~12 nm) indicates that the titanium modified this samples in order to increase the closed packing structure and hence decrease the pore size. In other words, Ti acts as lubricant in the lattice of ferrite to increase the closed packed structure. Figure 7 shows a direct relationship between the crystallite size obtained from XRD analysis and the pore size obtained from BET analysis. It confirms the above conclusion that as the crystallite size decrease, the closed packing of the sample increases, and hence the intestinal pores will decrease in size. This could be explained by assuming that the Ti is incorporated into ferrite structure in two positions: one in the framework and the other as inclusion $TiO_2$ particles. The framework sites make lubrication as we mentioned before. However, the inclusion of $TiO_2$ particles will make defects in the crystals and increase the pore size. These two factors reflect themselves in the previous curve by nonlinear relation between pore size and the crystallite size, although it is still direct relation. The role of surface area by this way will be neglected in the degradation of MB dye, hence the main role will be attributed to the electronic surface modification of the surface.

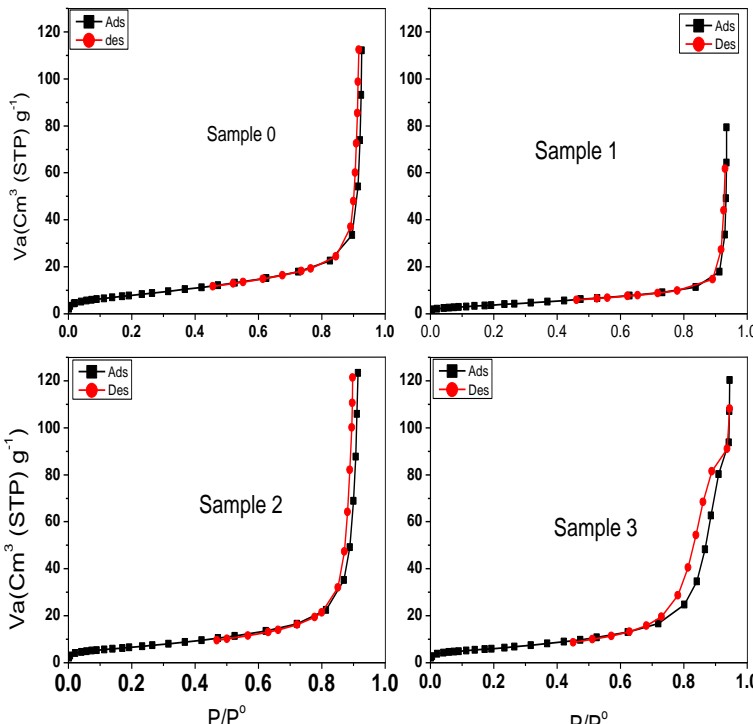

**Figure 6.** BET $N_2$ adsorption isotherms of all investigated samples.

**Table 4.** Surface area and pore size values of samples.

| Sample | Surface Area (m$^2$/g) | Pore Size (nm) |
|:---:|:---:|:---:|
| **0** | 30.1 | 17.1 |
| **1** | 15.39 | 19.8 |
| **2** | 25.3 | 15.9 |
| **3** | 23.27 | 12.8 |

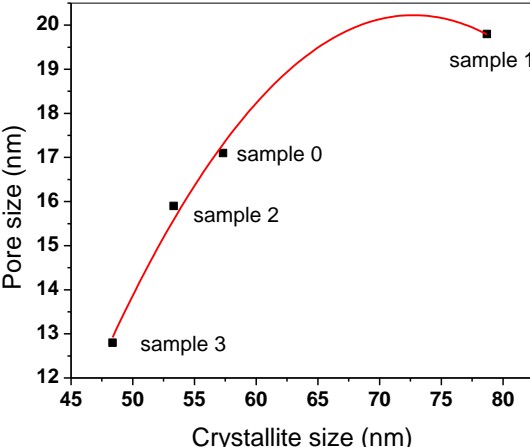

**Figure 7.** A correlation of crystallite and pore sizes.

*2.5. Photo Catalytic Degradation of Methylene Blue*

Figure 8 shows the photo degradation curves of all samples under UV irradiation and commercial white LED irradiation.

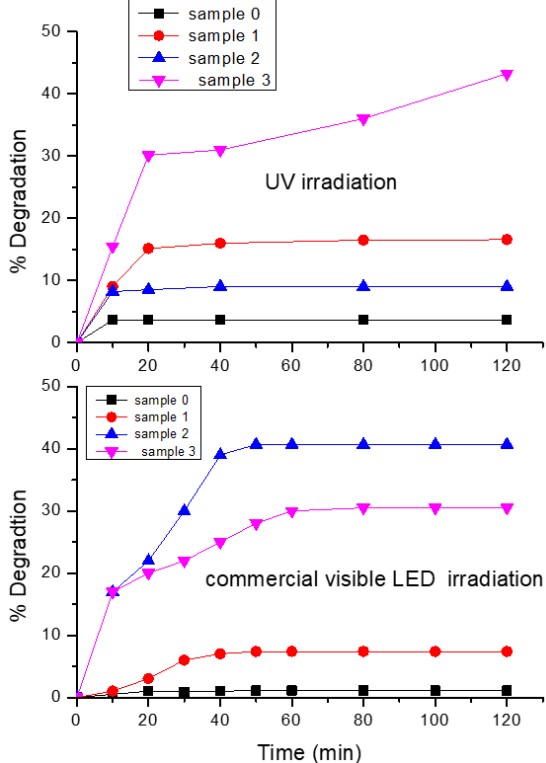

**Figure 8.** Photo degradation curves of methylene blue dye over different investigated samples under UV irradiation and commercial LED visible irradiation.

From this figure, it can be concluded that the undoped zinc ferrite showed very low photocatalytic activity in both irradiations.

Regarding to the commercial LED irradiation, sample 2 showed the highest photo degradation (~40%). Keeping this in mind, the previous observations are that the doping with Ti makes a modification of the surface, replacing Zn in lattice of zinc ferrite; in addition, as the % of Ti increases, this change increases until reaching maximum; after which, the tendency of Ti to form nano $TiO_2$ particles is enhanced rather than modifying the surface. This may be explained by the competition behavior of ions. In other words, when Ti ions are in a relatively small amount, the tendency to modify the surface is increased. In contrast, when it finds its similar ions in the neighbor, it accumulates and prefers to form nano $TiO_2$ particles rather than modification. It could be concluded that, in sample 2, modification of surface Zn reaches maximum so that sample 2 is the highest. Logically, we could conclude that sample 3 will be the second, which is clearly observed. However, sample 1, in which $TiO_2$ is proved to accumulate as nano particles, could be the lowest one, which is completely true, as can be seen from the figure.

In UV irradiation, sample 3, with the lowest amount of doping, showed that maximum degradation after sample 1 and not 2, which could be due to the presence of $TiO_2$ nano particles formed in this sample, and is active only in UV irradiation, after which sample 2 becomes the next; and also, sample 0 showed the minimum.

From the above data, it could be concluded that the doping with Ti makes surface modification of the surface by three aspects:

1. Substitute Zn in tetrahedral positions by Ti showed more UV response.
2. Rebelling off Zn forming nano zinc oxide, which is very active for visible irradiation.
3. Higher loading of Ti, resulting in formation of agglomeration of $TiO_2$ nanoparticles inside the crystals of zinc ferrite, which is also a UV active response.
4. The evidences are arisen that for both UV and LED irradiation, there is more than one active center existing.

*2.6. Kinetic Study of All Samples*

From Table 5 and analysis, it can be observed that the main fitted curves are the pseudo first order ones in both UV and visible irradiations. This is based on the value of regression factors. However, based on these values, we could observe that in sample 1, the value suggested that it is more fitted to pseudo second order (value of 0.998). Although, in visible irradiation, the highest values of regression factors of all samples suggested pseudo first order; however, for samples 2 and 3, high values are observed for second order.

**Table 5.** Shows the kinetic analysis of all samples in UV and commercial visible LED irradiations.

| | UV Irradiation | | | | | |
| --- | --- | --- | --- | --- | --- | --- |
| **Sample** | **1st Order** | | **2nd Order** | | **3rd Order** | |
| | **k** | **R²** | **k** | **R²** | **k** | **R²** |
| Sample 0 | 0.13 | 0.99 | 0.0355 | 0.519 | 0.0245 | 0.51 |
| Sample 1 | 0.224 | 0.97 | 0.0194 | 0.998 | 0.00581 | 0.75 |
| Sample 2 | 0.216 | 0.96 | 0.0042 | 0.566 | 0.00127 | 0.4122 |
| Sample3 | 0.28 | 0.94 | 0.0047 | 0.907 | 0.0006 | 0.808 |
| | Visible LED Irradiation | | | | | |
| **Sample** | **1st Order** | | **2nd Order** | | **3rd Order** | |
| | **k** | **R²** | **k** | **R²** | **k** | **R²** |
| Sample 0 | 0.04 | 0.957 | 0.0838 | 0.87 | 0.6208 | 0.775 |
| Sample 1 | 0.305 | 0.964 | 0.1378 | 0.80 | 0.3713 | 0.65 |
| Sample 2 | 0.32 | 0.974 | 0.0011 | 0.961 | 0.0002 | 0.918 |
| Sample 3 | 0.31 | 0.972 | 0.0011 | 0.957 | 0.0001 | 0.9389 |

The second order fitted curves means that two molecules of dye are degraded at once. In other words, this means that in these samples, we have more than one photo-active site. Thus, in sample 1 in UV irradiation, there are agglomerates of nano $TiO_2$ particles as discussed before in addition to the modified surface sites.

Moreover, in samples 2 and 3, in visible irradiation, the higher values in second order fitted curves suggested that we have also more than one visible photo-active site. This may be due to ZnO repelled phases and the modified surface sites.

We could conclude from this that the modified surface active sites are, more or less, active in both UV and visible irradiations. However, their activity is not equal in both irradiations.

## 3. Experimental

### 3.1. Materials

Zinc sulfate ($ZnSO_4 \cdot 7 H_2O$ (Merck, Rahway, NJ, USA)), iron nitrate ($Fe(NO_3)_3 \cdot 9H_2O$) (Merck), sodium hydroxide (NaOH) (Merck), titanium isopropoxide (Merck) (d = 0.955), and methylene blue dye (Merck).

### 3.2. Preparation of Zinc Ferrite and Doped Samples

The preparation was conducted using a co-precipitation method using sodium hydroxide as a precipitating agent. The titanium isopropoxide is added during the precipitation process. Then, samples are filtered, dried, and calcined at 700 °C for 2 h. The compositions of all samples prepared are summarized in the following Table 6, which shows the names of samples and the mole % ratios of each element in the prepared samples.

**Table 6.** Molar compositions of all prepared samples.

| | Mole % | | |
|---|---|---|---|
| **Sample** | **Ti** | **Zn** | **Fe** |
| 0 | 0 | 14.28 | 28.57 |
| 1 | 1.75 | 13.53 | 27.06 |
| 2 | 1.12 | 13.8 | 27.61 |
| 3 | 0.685 | 13.99 | 27.98 |

### 3.3. X-ray Powder Diffraction Analysis (XRD)

The crystalline phase and crystallite size of all catalyst nanoparticle samples were analyzed by X-ray powder diffraction (XRD) measurements, which were carried out at a Rigaku X-ray diffractometer system equipped with a RINT 2000 wide angle goniometer using Cu K$\alpha$ radiation ($\lambda$ = 0.15478 nm) and a power of 40 kV × 30 mA. The intensity data were collected at room temperature over a 2$\theta$ ranging from 10 to 80°. In addition, analysis of the crystal lattice analysis was performed with accompanied software that enables us to calculate the deviation in lattice parameters due to the incorporation of Ti.

### 3.4. Scanning Electron Microscopy (SEM)

The SEM technique is another important apparatus for morphological characterization of microporous and mesoporous molecular sieve materials. The micrographs obtained show the particle morphology (circular, cubic, etc.) of the synthesized materials, as well as the presence of any amorphous phase in the samples and the surface morphology and shape of particles of all catalyst nanoparticles. Samples were examined via a field emission scanning electron microscope (SEM), which was obtained using JEOL JSM-7600F. This system is combined with energy dispersive X-ray spectroscopy for composition and elemental analysis. With this tool, we could analyze the morphological structure of ferrite in addition to elemental analysis, enabling us to perform surface excess concentrations calculations.

### 3.5. Surface Area Measurements (Brunauer Emmett Teller (BET))

The textural properties (BET surface area, pore size distribution and pore volume) of the samples were obtained from N2-physisorption measurements at $-197\,^\circ$C. The Quantachrome ASiQ automated gas adsorption system was used to obtain the $N_2$ adsorption and desorption values for all the samples, in addition to performing statistical analysis to obtain the pore size distribution.

### 3.6. Photochemical Reactor

The instrument is handmade with three UV lamps type C 9 watt (200–280 nm) and two commercial LED strips (1500 lumens per meter) (400–700 nm). Each strip is one meter long (Figure 9).

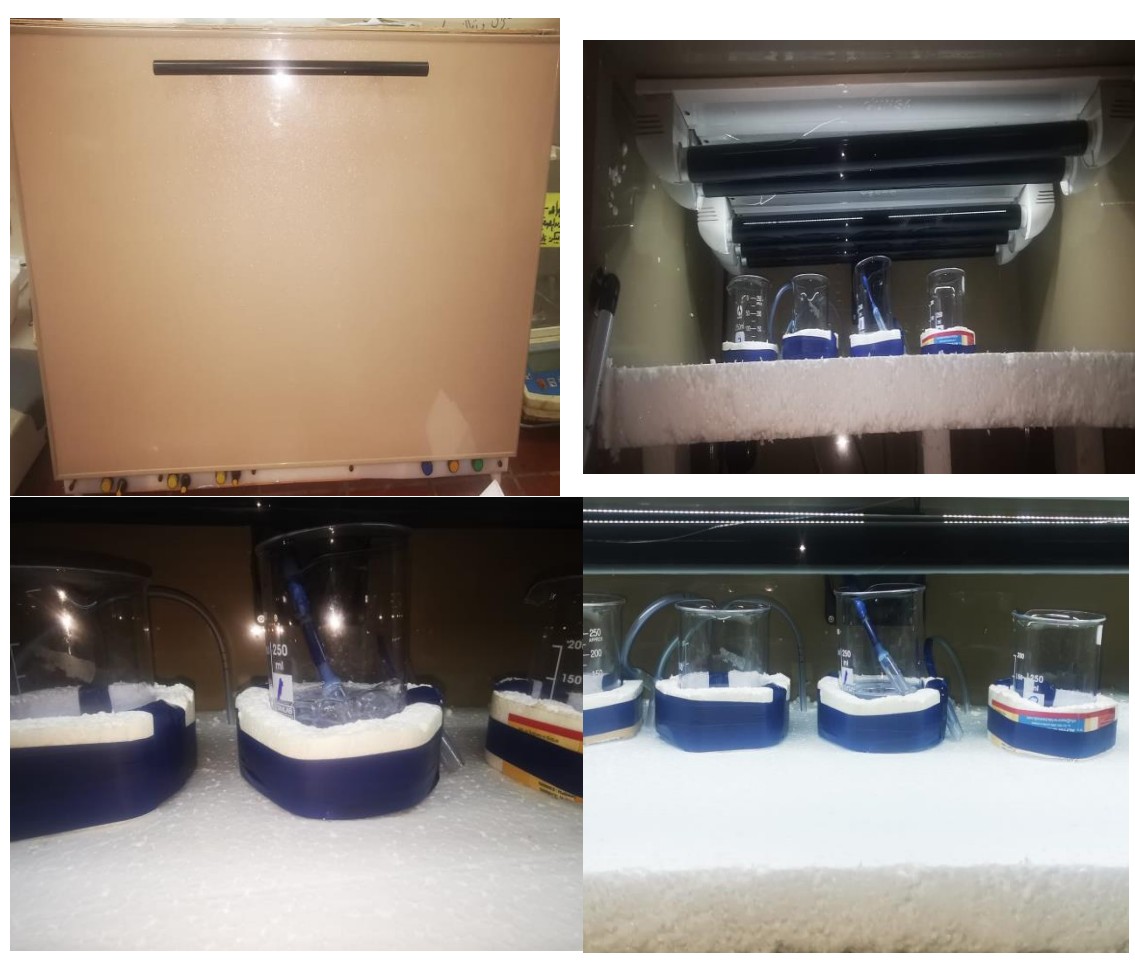

**Figure 9.** Photo image of instrument designed for measuring photo degradation.

The stirring is performed by passing air through a small air pump with four outlets. Catalyst is put with 100 mL of 10 PPM methylene blue dye.

Adsorption experiments were performed before irradiation by stirring with catalyst for 30 min and showed in all samples only about 1–4% adsorption. The data after adsorption were normalized to the new concentration to give the degradation %.

The photo images of the instrument are given here.

A small volume sample was taken every time interval for UV analysis.

DRUV–Vis Spectral Data

DRUV–vis spectral data were obtained using a UV-Vis spectrometer Jasco v770 (JASCO International Co., Tokyo, Japan) equipped with an integrating sphere in the wavelength range of 200–2000 nm to measure the reflectance spectra of samples.

## 4. Mechanism of Photo Degradation over Zinc Ferrite

Scheme 2 showed the proposed mechanism of photo degradation via low Ti doping, where two active centers are generated and one is tetrahedral Ti substituting zinc in the spinel structure of ferrite, and this is supposed to be UV active. The other active site is the rebelled zinc oxide, which is supposed to be active in visible irradiation.

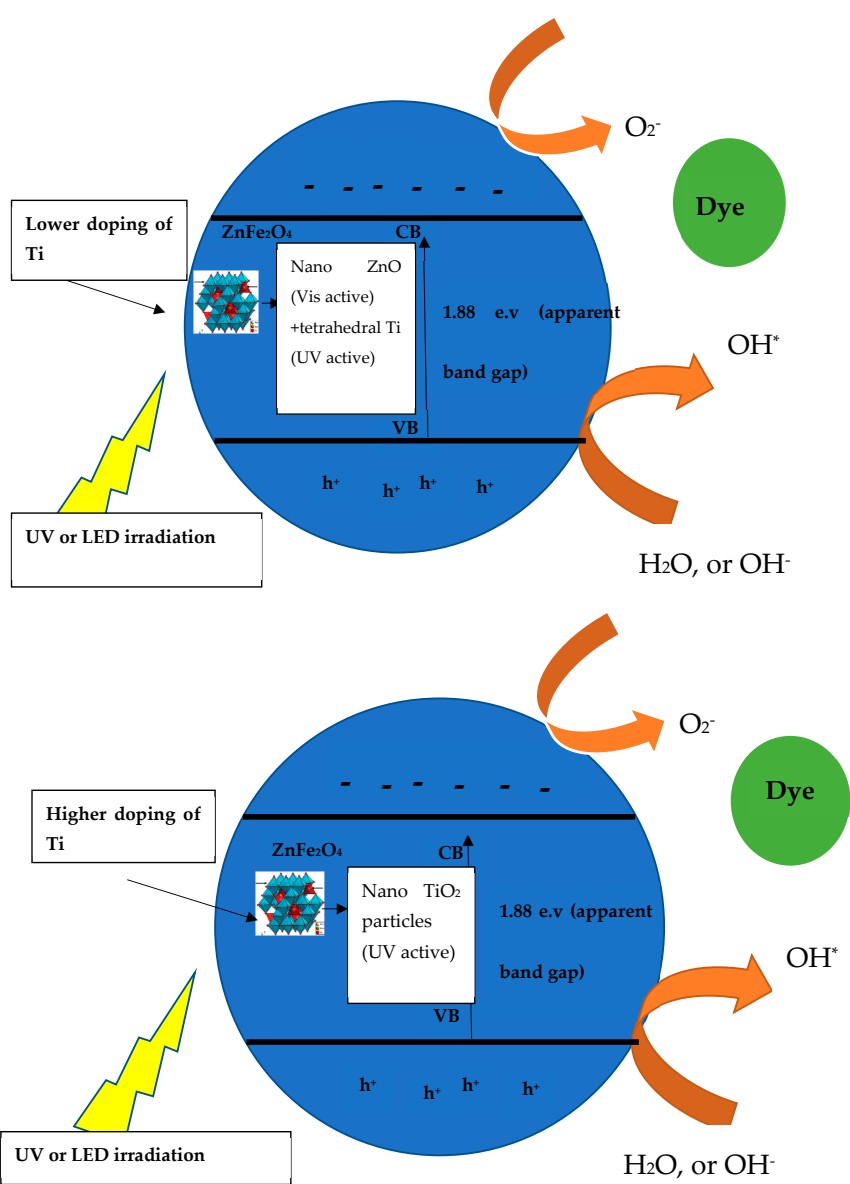

**Scheme 2.** Particles embedded in ferrite matrix. (* abbreviated to free radical).

On the other hand, large doping with Ti only generates agglomerates of nano particles of TiO$_2$, which are only active for UV irradiation.

The proposed mechanism showed that the irradiation transfers the electron from valence band to conduction band, resulting in generation of a OH$^*$ radical, which enables degradation of C-C and C-H bonds in the dye structure, resulting in complete decomposition of the dye. In addition, the bandgaps of the ferrite could be classified as apparent band gap and internal band gap. The apparent band gap is what could be measured directly through the diffuse reflectance measurements and attributed to the bulk average values of the band gap, while the internal band gap could be attributed to the fine change in the structure and generation of quantum levels due to these modifications. The internal

band gap could be only be detected as an active sites for photo degradation under LED commercial light.

1. The kinetic analysis showed that the degradation process follows up pseudo first-order reaction; however, some samples showed second-order ones, which enables us to conclude that there is more than one photo-active site.
2. The surface modified sites are photo-active in both UV and visible irradiations.
3. The surface area measurements showed a negligible role in photo degradation and the main role was found to be due to the electronic surface modification.

**Author Contributions:** Conceptualization, D.F.B. and I.H.A.E.M.; methodology, D.F.B. and I.H.A.E.M. investigation, D.F.B. and I.H.A.E.M.; resources, D.F.B. and I.H.A.E.M.; data curation, D.F.B. and I.H.A.E.M.; writing—original draft preparation, D.F.B. and I.H.A.E.M.; writing—review and editing, I.H.A.E.M.; visualization, I.H.A.E.M.; supervision, I.H.A.E.M.; project administration, D.F.B.; funding acquisition, D.F.B. All authors have read and agreed to the published version of the manuscript.

**Funding:** The Deanship of Scientific Research (DSR) at King Abdulaziz University (KAU), Jeddah, Saudi Arabia has Funded this project under grant no (G:606-247-1443).

**Data Availability Statement:** Not applicable.

**Conflicts of Interest:** The authors declare no conflict of interest.

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
