# Peer review of "Surface Modification of Zinc Ferrite with Titanium to Be a Photo-Active Catalyst in Commercial LED Light"

_catalysts, doi:10.3390/catal13071082_

Round 1

Reviewer 1 Report

This paper synthesized Ti doped zinc ferrite and investigated its photocatalytic activity in MB degradation. The improved activity was observed and explained. However, some serious issues exist. In the current state, it is not suitable for publication.

1、 The photocatalytic performance of the Ti doped zinc ferrite is very low. Given the poor photocatalytic activity of the samples, the novelty of the work is limited.

2、 Why the samples with the best performance under UV light and LED light are not the same, the authors should explain in depth.

3、 The adsorption experimental data before the degradation experiment need to be provided.

4、 The definition of SEM images is too low, and the author needs to provide images with higher magnification.

5、Some pictures are simple and rough.

1There are some spelling mistakes, such as the ordinate of Figure 8 is misspelled.

2Many sentences are too wordy and have grammatical errors. For instance,Although the ferrite materials have been used as catalyst for many reactions however the fact that very limited application for its use as a photo catalyst is a fact.should be “Although ferrite materials have been used as catalysts for many reactions, their applications as photocatalysts are actually very limited.”

Author Response

First of all , We would like to express deep appreciations to the reviewer for his valuable comments which are  really raise the level of our paper. here are the replying point by point 

This paper synthesized Ti doped zinc ferrite and investigated its photocatalytic activity in MB degradation. The improved activity was observed and explained. However, some serious issues exist. In the current state, it is not suitable for publication.

1、 The photocatalytic performance of the Ti doped zinc ferrite is very low. Given the poor photocatalytic activity of the samples, the novelty of the work is limited.

Although the photo catalytic activity of the doped samples  did not exceed 40% degradation we suggest that the raising of photocatalytic activity of zinc ferrite from nearly null to this value is considered to be a  good step for achieving  much performance in future work. In addition the activity of the doped materials in commercial LED light attain more  and more interesting results to use these materials  in ordinary LED light which is considered to be a good results  with respect to this kind of irradiation comparing other high price catalysts.

2、 Why the samples with the best performance under UV light and LED light are not the same, the authors should explain in depth.

This is because we suggest that there is more than one active center in the prepared samples each active center showed different behavior towards different kinds of irradiation and this is due to the presence of different quantum levels in the prepared materials enables it to transfer electrons through  different band gaps rather than the apparent average measured band gap

3、 The adsorption experimental data before the degradation experiment need to be provided.

This already done from beginning and the data of this part is provided in the revised version.

4、 The definition of SEM images is too low, and the author needs to provide images with higher magnification.

The image of SEM is at 100000 magnifications and the images are improved in the revised version however trying to increase magnification give hazier images due to deviation of electron beam due to highly magnetization of ferrite within the capability of our instrument.

5、Some pictures are simple and rough.

All images are  improved in the revised version

Comments on the Quality of English Language

1、There are some spelling mistakes, such as the ordinate of Figure 8 is misspelled.

This is corrected

2、Many sentences are too wordy and have grammatical errors. For instance,“Although the ferrite materials have been used as catalyst for many reactions however the fact that very limited application for its use as a photo catalyst is a fact.”should be “Although ferrite materials have been used as catalysts for many reactions, their applications as photocatalysts are actually very limited.”

This  is corrected in the revised version.

Reviewer 2 Report

This manuscript explored the synthesis method and photocatalytic activity of Ti doped ferrite. The authors tried different concentrations of Ti and concluded that the ferrite was modified by different methods varying from the ratios of doped Ti. The doped catalysts show increasing activities of photo degradation, comparing with the bare ferrite. The catalyst was characterized by XRD, SEM, EDX, BET, UV-VIS/DRS, etc. The manuscript realized the conversion of photo-inactive zinc ferrites into a promising photocatalyst by Titanium doping. Some critical issues should be addressed.

1. The quality of SEM images is poor and should be improved.

2. In the sample preparation part, why did the authors choose the ratios showed in the chart? How to decide the synthesis schedule?

3. There is no need to display the ordinate in XRD pattern.

4. What is the condition of valence state in the compounds? Please provide the XPS spectra.

5. The evidence of the conclusion “Ti substituting Zinc” is not sufficient. Providing the HRTEM images or other evidence would be better.

6. The scheme 2 doesn’t explain the conclusion clearly and should be replaced.

7. In the SEM section, the authors claimed that the result of EDX shows the Ti accumulates on the surface in low doped amount (samples 2 & 3) while the Ti tends to enter the crystal of zinc ferrite as inclusion as TiO2 in larger doped amount (sample 1). Can the authors give an explanation of the phenomenon?

no

Author Response

First of all , We would like to express deep appreciations to the reviewer for his valuable comments which are  really raise the level of our paper. here are the replying point by point 

2nd reviewer

This manuscript explored the synthesis method and photocatalytic activity of Ti doped ferrite. The authors tried different concentrations of Ti and concluded that the ferrite was modified by different methods varying from the ratios of doped Ti. The doped catalysts show increasing activities of photo degradation, comparing with the bare ferrite. The catalyst was characterized by XRD, SEM, EDX, BET, UV-VIS/DRS, etc. The manuscript realized the conversion of photo-inactive zinc ferrites into a promising photocatalyst by Titanium doping. Some critical issues should be addressed.

  1. The quality of SEM images is poor and should be improved.

This is improved in revised version

  1. In the sample preparation part, why did the authors choose the ratios showed in the chart? How to decide the synthesis schedule?

These ratio are chosen according to the molar ratio in the scheme we begin with very low amount in order to not lose the magnetization behavior of ferrite

  1. There is no need to display the ordinate in XRD pattern.

The ordinate is removed

  1. What is the condition of valence state in the compounds? Please provide the XPS spectra.

Unfortunately XPS instrument is not available at this time in our institute. However , the valence state of Zinc and Iron are well known to be Zn2+, and Fe3+ in ferrite structure or in ZnO phases. The Ti should exhibit +4 in bulk which is confirmed from DR spectra. XPS spectra  is expected to show only the surface ions which may maintain Ti3+ if it available.

  1. The evidence of the conclusion “Ti substituting Zinc” is not sufficient. Providing the HRTEM images or other evidence would be better.

Ti substituting Zinc is evidenced in our paper by two evidences; one from XRD lattice analysis and the other from deconvolution analysis of DR. Although the HRTEM could give more details however the magnetization of our samples prevents from focusing to high magnification which gives only hazy images.

  1. The scheme 2 doesn’t explain the conclusion clearly and should be replaced.

This scheme is improved

  1. In the SEM section, the authors claimed that the result of EDX shows the Ti accumulates on the surface in low doped amount (samples 2 & 3) while the Ti tends to enter the crystal of zinc ferrite as inclusion as TiO2in larger doped amount (sample 1). Can the authors give an explanation of the phenomenon?

This phenomena is regarded to the competition between components when the Ti is exist in low amount it tends to make a substitution to Zinc ions while when Ti exists in larger amount it tends  to accumulate on its own other Ti particles to  accumulate. And this is mentioned in the revised version.

Round 2

Reviewer 1 Report

 The authors did a great job on revision. I have no further comments on the manuscript.